# Prospective Cohort Study of Sociodemographic and Work-Related Factors and Subsequent Unemployment under COVID-19 Pandemic

**DOI:** 10.3390/ijerph19116924

**Published:** 2022-06-06

**Authors:** Makiko Kuroishi, Tomohisa Nagata, Ayako Hino, Seiichiro Tateishi, Akira Ogami, Mayumi Tsuji, Shinya Matsuda, Koji Mori, Yoshihisa Fujino

**Affiliations:** 1Department of Occupational Health Practice and Management, Institute of Industrial Ecological Sciences, University of Occupational and Environmental Health, Kitakyushu 807-8555, Japan; mim.kuro@gmail.com (M.K.); kmori@med.uoeh-u.ac.jp (K.M.); 2Department of Mental Health, Institute of Industrial Ecological Sciences, University of Occupational and Environmental Health, Kitakyushu 807-8555, Japan; ayako-hino@med.uoeh-u.ac.jp; 3Disaster Occupational Health Center, Institute of Industrial Ecological Sciences, University of Occupational and Environmental Health, Kitakyushu 807-8555, Japan; tateishi@med.uoeh-u.ac.jp; 4Department of Work Systems and Health, Institute of Industrial Ecological Sciences, University of Occupational and Environmental Health, Kitakyushu 807-8555, Japan; gamisan@med.uoeh-u.ac.jp; 5Department of Environmental Health, School of Medicine, University of Occupational and Environmental Health, Kitakyushu 807-8555, Japan; tsuji@med.uoeh-u.ac.jp; 6Department of Preventive Medicine and Community Health, School of Medicine, University of Occupational and Environmental Health, Kitakyushu 807-8555, Japan; smatsuda@med.uoeh-u.ac.jp; 7Department of Environmental Epidemiology, Institute of Industrial Ecological Sciences, University of Occupational and Environmental Health, Kitakyushu 807-8555, Japan; zenq@med.uoeh-u.ac.jp

**Keywords:** sociodemographic factors, socioeconomic status, unemployment, COVID-19, Japan

## Abstract

The previous studies found that women and low-income households were more likely to experience unemployment prior to the COVID-19 pandemic. However, there is no cohort study to examine the relationship during the COVID-19 pandemic. The aim of this prospective cohort study is to examine the relationship between sociodemographic factors and unemployment during the COVID-19 pandemic in Japan. We surveyed the socioeconomic status, personal characteristics, and occupation of recruited workers at baseline (22–25 December 2020); subsequent unemployment was examined at follow-up (18–19 February 2021). We determined the odds ratio of unemployment by sociodemographic status and occupation. The multivariate model was adjusted for sex and age. Among the 19,941 participants, 725 (3.6%) had experienced unemployment. Multivariate analysis showed significant high unemployment amongst women and participants of younger age, bereaved or divorced, unmarried, of lower income, or with short educational background. By occupation, the unemployment rate of temporary or contract employees and self-employed is high. COVID-19 expelled socially vulnerable groups from employment. This suggests the need for employment and economic support for such individuals.

## 1. Introduction

Unemployment is associated with a substantial risk of poor physical and mental health. It has consistently been shown to be significantly associated with increases in chronic heart disease, acute myocardial infarction, poor mental health, mental disorders, substance-related disorders, and suicide [1,2,3]. Unemployment is an important factor with regard to public health: it poses a health risk for individuals; it also leads to family poverty and, ultimately, constitutes a burden on social security.

COVID-19 has spread around the world and continues to exert a profound impact on global economies. Economic activities have changed drastically as a result: people have refrained from travel and outside eating, drinking, and entertainment; they are encouraged to stay at home. The resulting unemployment has become a major concern: according to the Organization for Economic Co-operation and Development (OECD), countries’ unemployment rates rose significantly around the same time as the outbreak of COVID-19 [4]. In this regard, Japan’s unemployment rate appears to have remained relatively low—around 3%. However, official data on the unemployment rate may be underestimated, as it is defined as “people looking for work”. Data for Japan clearly show a marked decline in the number of people in employment and a fall in household income [5,6].

With disasters and crises in the past, the risk of unemployment was found to be particularly high among socially vulnerable groups [7]. The OECD has identified young people, women, middle-aged and older individuals, and migrants as vulnerable groups in the labor market [8]. Those workers are also reported to be more likely to leave the labor force owing to unemployment, disability, or economic inactivity [9]. By contrast, civil servants, teachers, and employees of non-profit organizations in Japan have been found to have lower turnover rates and greater job security [10].

It is assumed that individual sociodemographic status is still significantly associated with unemployment during COVID-19. A cross-sectional study conducted in the United States examined adverse outcomes associated with COVID-19 and the country’s stay-home policies. It found that African Americans, Hispanics, women, and low-income households were more likely to experience unemployment, food insecurity, mental health problems, poor access to health care, and rent or mortgage delinquency [11]. However, there is no prospective cohort study of the relationship between sociodemographic status and unemployment among the Japanese during COVID-19, which is unclear.

It is important that we conducted a prospective cohort study in Japan. First, evidence from longitudinal studies regarding the COVID-19 pandemic does not exist. Second, Japan has never in the past taken measures equivalent to a blockade, even in the case of a new strain of influenza. This is the first time that strong social measures have been taken to combat infectious diseases, and it is unclear what factors are associated with unemployment under the pandemic in Japan.

The hypothesis is that the economically vulnerable are presumed to be more likely to be unemployed. The purpose of this study is to examine the association between the sociodemographic factors of workers and unemployment by means of a prospective cohort during the COVID-19 pandemic in Japan.

## 2. Materials and Methods

The Methods section describes the study design, data collection and statistical analysis.

### 2.1. Study Design

This prospective cohort study about COVID-19 among Japanese workers was conducted under the Collaborative Online Research on the Novel-coronavirus and Work (CORoNaWork) Project. Details of the study protocol are described elsewhere [12]. Briefly, we administered a baseline questionnaire on 22–25 December 2020 and a follow-up questionnaire on 18–19 February 2021, when Japan was in its third pandemic wave. This pandemic wave was larger than the previous first and second waves in Japan.

For the baseline survey, we recruited 33,087 workers throughout Japan from 605,381 randomly selected panelists registered with an Internet survey company. The inclusion criteria for participants were being currently employed and aged 20–65 years. Survey items included basic socio-demographic characteristics such as family structure, income, education, place of residence, place of work, and work environment, with 43 questions in the first round and 23 questions in the second round [12]. We applied cluster sampling with stratification by sex, job type, and region. We excluded 6051 invalid responses owing to the following: response time < 6 min; body weight < 30 kg; height < 140 cm; inconsistent answers to similar questions; and incorrect answers to questions intended to identify fraudulent responses. We distributed the follow-up questionnaire to the 27,036 people with valid responses to the baseline questionnaire. In total, 19,941 participants completed both questionnaires (follow-up rate, 73.8%). Among the participants, the combined number of close contacts and infected persons was 295 (1.5%). The specifics of the questionnaire are provided in Appendix A.

This study was approved by the Ethics Committee of the University of Occupational and Environmental Health, Japan (reference nos. R2-079 and R3-006). Informed consent was obtained from all participants.

### 2.2. Data Collection

We retrieved the following data from the baseline survey for inclusion as explanatory variables: age; sex; marital status; sociodemographic status (based on annual household income and education); occupation; and job type. We categorized age into the following five groups: 20–29; 30–39; 40–49; 50–59; and ≥60 years. Marital status was classified into four groups: married (working spouse); married (spouse not working); divorced or widowed; and never married. Annual household income was classified into the following six groups: under 2 million; 2–4 million; 4–6 million; 6–8 million; 8–10 million; and >10 million yen. We categorized education into five groups: up to junior high school; up to high school; up to junior college or technical school; up to university; and graduate school. We categorized occupation into seven groups: general employee; manager; executive manager; public employee, faculty member, or non-profit organization employee; temporary or contract employee; self-employed; and other. Job type was classified into three categories: mainly desk work; work mainly involving interpersonal communication; and mainly manual or physical labor.

We ascertained unemployment as follows. First, the baseline survey included only people who were employed at the time of response. In the follow-up survey, in answer to the question “Have you changed your place of work since December 2020?” respondents were asked to select one of the following six options: “no change”; “I was transferred to another company”; “I resigned and got a new job right away”; “I stopped working and was unemployed for a while but am now working”; “I stopped working and started a business (e.g., managing a company, running a sole proprietorship, or engaging in self-employment)”; and “I stopped working and am not currently working (including job seeking)”. We defined unemployment as participants choosing one of the following: “I resigned and got a new job right away”; “I stopped working and was unemployed for a while but am now working”; “I stopped working and started a business (e.g., managing a company, running a sole proprietorship, or engaging in self-employment)”; or “I stopped working and am not currently working (including job seeking)”.

### 2.3. Statistical Analysis

We determined the odds ratios (ORs) of unemployment for sociodemographic status and occupation using a multilevel logistic model for the prefecture of residence. The multivariate model was adjusted for sex and age. We undertook a trend test by conducting the analysis using age, annual household income, and education as continuous variables. We also used the incidence rate of COVID-19 by prefecture as a prefecture-level variable. The multilevel analysis was performed nested by the incidence rate of COVID-19. We considered *p* values under 0.05 statistically significant. All analyses were conducted using Stata (Stata Statistical Software release SE16.1; StataCorp LLC, College Station, TX, USA).

## 3. Results

The Results section describes the characteristics of the participants and sociodemographic factors and unemployment.

### 3.1. The Characteristics of the Participants

The characteristics of the respondents appear in Table 1. There were 11,170 men in the sample, accounting for 56% of the total. The mean age was 48.0 years. In total, married workers were 11,185 (66%) and 8880 (45%) workers had graduated from university. The majority of occupation were general employee (46%) and that of job type were desk workers (52%).

### 3.2. Sociodemographic Factors and Unemployment

Table 2 shows the associations of sociodemographic status (including occupation) with unemployment: 725 (3.6%) workers had experienced unemployment.

Multivariate analysis showed that the OR of unemployment associated with sex was 1.35 (95% confidence interval [CI], 1.14–1.60) for women compared with men. With increasing age, the OR for unemployment was lower: OR, 0.98; 95% CI, 0.97–0.99; *p* < 0.001, adjusted for sex. The respective OR and 95% CI figures for the association with marital status were as follows: 1.33 (1.03–1.71) for being married (spouse not working); 2.09 (1.65–2.64) for being bereaved or divorced; and 1.29 (1.07–1.56) for being unmarried, compared with being married (spouse working). The respective figures for the association with annual household income were as follows: 4.05 (3.00–5.46) for <2 million yen; 2.12 (1.62–2.78) for 2–4 million yen; and 1.46 (1.11–1.93) for 4–6 million yen, compared with >10 million yen. The respective figures for the association with education were as follows: 1.73 (1.12–2.66) for junior high or high school and 1.83 (1.19–2.83) for vocational school, junior college, or technical school. The respective figures for the association with occupation were as follows: 2.01 (1.63–2.48) for temporary or contract employees and 1.35 (1.02–1.78) for being self-employed, compared with general employees, while the figures were 0.56 (0.40–0.79) for public employees, faculty members, or non-profit organization employees. The respective figures for the association with job type were 1.25 (1.04–1.51) for jobs mainly involving interpersonal communication and 1.85 (1.55–2.21) for mainly manual or physical labor, compared with mainly desk work.

## 4. Discussion

Through a cohort study, this investigation examined the association between sociodemographic factors and subsequent unemployment during COVID-19. Under the pandemic of COVID-19, we found that unemployment was associated with sociodemographic factors such as age, sex, marriage, income, education, and occupation.

We observed that the risk of unemployment was highest among young people. International Labour Organization (ILO) has reported that worldwide, young people were subjected to the greatest loss of labor opportunities through COVID-19: in 2020, young workers suffered 9.1% job losses compared with 2.6% for adults [13]. The present study conducted in Japan similarly found that young people were more than twice as likely to be unemployed as middle-aged and older workers. This could have been due to the fact that the accommodation and service industry as well as the food and beverage industry (which were most affected by COVID-19) had more young casual workers than other industries [5,14]. In addition, even before the pandemic, there was a high turnover rate of young people in Japan. The turnover rate within 3 years of graduating from university and entering the workforce was around 30% in 2021 [15]. This is thought to be due to their transition from school to work and still looking for the right job.

By occupation and job type, we found the unemployment rate to be higher among temporary and contract workers and manual laborers. Japan’s Labour Force Survey reported that the number of non-regular workers in the country fell sharply under COVID-19 [5]. Many temporary and contract workers were employed in lifestyle-related industries, travel, and entertainment services, which were heavily affected by the pandemic [16,17]. Temporary and contract workers are thought to be used to adjust employment and our results clearly confirm this.

We found that women were more likely to be unemployed than men. Prior to the COVID-19 pandemic, the OECD has stated that women are also more vulnerable in society [8]. On the other hand, under the COVID-19 pandemic, a previous study revealed that women were less likely than men to leave their jobs [11]. That previous study calculated odds ratios adjusted for age, education, marital status, and number of individuals in the household, whereas our study is an odds ratio adjusted for age only. In our present study, we found that 14.7% of women (compared with 7.4% of men) were in informal employment, and this may have led to the tendency for women to leave the workforce. Japan’s Labour Force Survey during COVID-19 observed a gender difference in the decline in the number of people in employment: women were more likely to be unemployed [5]. The 2021 survey reported that 68% of people in informal employment were women; many of them worked in the accommodation, catering, and lifestyle-related service industries [5].

The unemployment rate was also significantly higher for divorced or bereaved people than with dual-earner households. In economic terms, marriage is held to be a rational behavior that seeks economic gain [18,19]. It has long been pointed out that low-income earners and those with unstable employment are less likely to get married [20,21]. The unmarried participants with their precarious employment situation may became unemployed owing to the pandemic. Moreover, if the divorced or bereaved had a child, it is possible that they were forced to leave the workplace due to school or after-school care leave, because of school closures related to the pandemic [22].

With regard to income and education, we observed that the lower the income and lower the education, the greater was the likelihood of unemployment. It is widely known that such socially vulnerable groups are at higher risk of unemployment [7,9,23]; we found a similar trend during COVID-19. The impact of unemployment on the lives of those with lower incomes is accordingly greater, and they constitute the group in highest need of social support.

Previous studies have shown that people of lower sociodemographic status are more likely to face difficulties in the event of a pandemic. However, much of the research has focused on the higher risk of contracting infectious diseases and, as a result, being more likely to face problems such as healthcare costs and unemployment [7,24,25,26,27]. Khanijahani et al. have issued a systematic review of health inequalities by COVID-19 on morbidity and mortality [28]. Socially vulnerable groups have higher morbidity and mortality. However, it is not only whether a person has an infectious disease that determines long-term prognosis. Social factors such as job availability and poverty also need to be taken into account. Our findings suggest that in the event of a major epidemic, resulting in unemployment among vulnerable segments of the labor market, regardless of whether workers themselves are infected. Thus, there is a need for employment and financial support for socially vulnerable groups in the event of a major epidemic.

### Limitations

There are several limitations of this study. First, it was unclear why the participants had experienced unemployment: we did not know whether it was due to the effects of COVID-19, company bankruptcy or financial difficulties, or the participants’ voluntary decision to change jobs. It has also been found that the COVID-19 pandemic is closely associated with firms’ business conditions, especially in small and medium-sized enterprises [29]. Future research using unemployment as an outcome should also take into account the business conditions of firms. Second, this study was conducted as an Internet survey; thus, the generalizability of our results is unclear. It is possible that individuals who were genuinely penurious did not have Internet access and could not participate in the survey. If such people had taken part in the survey, the bias would have been stronger. We attempted to reduce subject bias as much as possible by sampling by region and occupation based on infection rates. Third, of the 27,036 individuals who participated in the baseline survey, 7095 did not respond to the follow-up survey (non-participation rate, 26%). It is possible that those who have left their jobs have not responded to the follow-up survey in this study. In that case, the results of this study may have reinforced the trend.

The survey was conducted in 2020 and 2021, and we plan to add follow-up surveys to analyze which attributes make people more likely to re-enter the workforce, as well as the impact on physical and mental health. Measures to reduce infectious disease cases and deaths were of course important, while it would be necessary to follow up closely to see whether the significant slowdown in economic activity had not worsened the health of the unemployed in the future and increased the excess mortality rate in the long term.

## 5. Conclusions

In the COVID-19 pandemic, the relationship between sociodemographic factors and subsequent unemployment was confirmed. Similar to previous studies prior to the COVID-19 pandemic, vulnerable groups were more likely to be unemployed. Lockdowns and other behavioral restrictions will be necessary to prevent infection, but they have the side effect of stagnating the economy. Unemployment causes various health disadvantages. To avoid an increase in long-term excess mortality due to unemployment as a ‘side effect’ of infectious disease control, it is necessary to provide widespread and sustained support to the high-risk groups identified in this case, in the form of short- and long-term vocational training and health care.

## Figures and Tables

**Table 1 ijerph-19-06924-t001:** The characteristics of the participants.

		N (%)
Number of subjects		19,941
Sex, Men		11,170 (56.0%)
Age		
	20–29	1055 (5.3%)
	30–39	3218 (16.1%)
	40–49	5929 (29.7%)
	50–59	7095 (35.6%)
	60–	2644 (13.3%)
Marital status		
	married (spouse is working)	8125 (40.7%)
	married (spouse is not working)	3060 (15.3%)
	bereaved/divorced	2036 (10.2%)
	unmarried	6720 (33.7%)
Annually household income (million JPY)		
	<2	1288 (6.5%)
	≥2 and <4	3989 (20.0%)
	≥4 and <6	4772 (23.9%)
	≥6 and <8	3967 (19.9%)
	≥8 and <10	2629 (13.2%)
	≥10	3296 (16.5%)
Education		
	Junior high or high school	5360 (26.9%)
	Vocational school, junior college or technical school	4585 (23.0%)
	University	8880 (44.5%)
	Graduate School	1116 (5.6%)
Occupation		
	general employee	9098 (45.6%)
	manager	2088 (10.5%)
	executive manager	691 (3.5%)
	public employee, faculty member, or non-profit organization employee	2010 (10.1%)
	temporary or contract employee	2109 (10.6%)
	self-employed	1737 (8.7%)
	others	2208 (11.1%)
Jobtype		
	mainly desk work	10,268 (51.5%)
	jobs mainly involving interpersonal communication	4937 (24.8%)
	mainly manual or physical labor	4736 (23.8%)

**Table 2 ijerph-19-06924-t002:** The association between sociodemographic, work-related factors, and unemployment.

		Total	Unemployment Rate	Non-Adjusted				Age-Sex-Adjusted *			
		N	%	OR	95% CI		*p* Value	OR	95% CI		*p* Value
Total		19,941	3.6								
Sex											
	Men	11,170	2.9	reference				reference			
	Women	8771	4.6	1.62	1.40	1.88	<0.001	1.35	1.14	1.60	<0.001
Age											
	20–29	1055	8.1	reference			<0.001 †	reference			<0.001 †
	30–39	3218	5.2	0.62	0.47	0.81	0.001	0.64	0.49	0.84	<0.001
	40–49	5929	3.1	0.36	0.28	0.47	<0.001	0.40	0.31	0.53	<0.001
	50–59	7095	2.9	0.33	0.26	0.43	<0.001	0.39	0.30	0.52	<0.001
	60–	2644	3.4	0.40	0.29	0.54	<0.001	0.49	0.35	0.67	<0.001
Marital status											
	married (spouse is working)	8125	2.8	reference				reference			
	married (spouse is not working)	3060	3.2	1.14	0.90	1.46	0.279	1.33	1.03	1.71	0.028
	bereaved/divorced	2036	5.5	2.03	1.61	2.56	<0.001	2.09	1.65	2.64	<0.001
	unmarried	6720	4.3	1.56	1.31	1.86	<0.001	1.29	1.07	1.56	0.007
Annually household income (million JPY)							<0.001 †				<0.001 †
	<2	1288	9.1	4.35	3.23	5.86	<0.001	4.05	3.00	5.46	<0.001
	≥2 and <4	3989	5.2	2.38	1.82	3.12	<0.001	2.12	1.62	2.78	<0.001
	≥4 and <6	4772	3.5	1.57	1.19	2.06	0.001	1.46	1.11	1.93	0.008
	≥6 and <8	3967	2.5	1.13	0.84	1.53	0.421	1.06	0.78	1.44	0.699
	≥8 and <10	2629	2.2	0.98	0.69	1.38	0.892	0.94	0.66	1.32	0.709
	≥10	3296	2.3	reference				reference			
Education							0.025 †				0.011 †
	Junior high or high school	5360	3.7	1.77	1.15	2.72	0.009	1.73	1.12	2.66	0.013
	Vocational school, junior college, or technical school	4585	4.3	2.07	1.35	3.19	0.001	1.83	1.19	2.83	0.006
	University	8880	3.5	1.64	1.08	2.50	0.021	1.50	0.98	2.28	0.060
	Graduate School	1116	2.2	reference				reference			
Occupation											
	general employee	9098	3.5	reference				reference			
	manager	2088	1.7	0.46	0.33	0.66	<0.001	0.59	0.41	0.85	0.005
	executive manager	691	2.3	0.65	0.39	1.08	0.096	0.84	0.50	1.40	0.499
	public employee, faculty member, or non-profit organization employee	2010	1.8	0.52	0.37	0.74	<0.001	0.56	0.40	0.79	0.001
	temporary or contract employee	2109	6.5	1.92	1.56	2.36	<0.001	2.01	1.63	2.48	<0.001
	self-employed	1737	3.9	1.10	0.84	1.44	0.472	1.35	1.02	1.78	0.035
	others	2208	5.1	1.47	1.18	1.84	<0.001	1.47	1.18	1.84	<0.001
Jobtype											
	mainly desk work	10,268	2.9	reference				reference			
	jobs mainly involving interpersonal communication	4937	3.8	1.34	1.11	1.62	0.002	1.25	1.04	1.51	0.018
	mainly manual or physical labor	4736	5.1	1.83	1.53	2.17	<0.001	1.85	1.55	2.21	<0.001

The multilevel analysis was performed nested by the incidence rate of COVID-19 by prefecture as a prefecture-level variable. * Sex category was calculated for adjusted by age, and age category was calculated for adjusted by sex. † *p* for trend.

## Data Availability

The data that support the findings of this study are available from the corresponding author upon reasonable request.

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
