# Peer review of "Prospective Cohort Study of Sociodemographic and Work-Related Factors and Subsequent Unemployment under COVID-19 Pandemic"

_ijerph, 2022, doi:10.3390/ijerph19116924_

Round 1

Reviewer 1 Report

The paper investigates the relationship between socio-demographic factors and unemployment during the COVID-19 pandemic in Japan. The findings of this paper are insteresting and have significant theoretical and policy implications.

  1. The paper claims that it is dedicated to examine the relationship between socioeconomic status and unemployment (page 2 line 69-70). It actually investigates the relationship between sociodemographic status and unemployment.
  2.  Since the survey is about the  unemployment under COVID-19, the author may consider to control whether respondents are affected or under risk of COVID-19 to identify if the unmployment is because of COVID-19.
  3. The author say that the paper uses the incidence rate of COVID-19 by prefecture as a prefecture-level variable. I cannot see where it has been used. Please make it clear.
  4. For the sociodemographic factors, the author may consider if respondents have children or not. This is related to the employment of adults, especially female.

Reviewer 2 Report

It is commendable that a large-scale survey on unemployment was conducted during COVID-19. However, there is nothing new in the conclusion. It should be "Major Revision" or "Reject". It's up to the editor to decide which one to apply.
What is the uniqueness of this study? OECD reports and US cases are covered, but are the analytical methods and results the same? If so, what is the value of this paper? Simply because it was done in Japan will not attract the attention of many readers. Relatedly, please explain why the author focused on Japan.
There is little discussion of previous research. A section on previous research should be set up for careful discussion. In addition, the hypotheses tested by this study should be clearly presented.
Is the expression "Baseline" in Table 1 common even for cross-sectional analysis like this paper?
Since this study is based on cross-sectional analysis, the following expressions are incorrect. "Overall, COVID-19 appears to have increased difficulties for a previously vulnerable group."
The conclusion reached is that the vulnerable are unemployed. Although the survey was conducted during the period of COVID-19, the influence of COVID-19 could not be shown. Authors should make comparisons using data from studies conducted prior to COVID-19. If this is not possible, at the very least, it should be compared with the results of previous studies and a careful discussion should be made as to what changes may have occurred during COVID-19.
In the limitation section, please add what you need for future research based on the limitation of this paper.

Reviewer 3 Report

Dear Authors
I want to congratulate you for this robust and interesting and pertinent scientific study related to a reality that is very important for everyone's life.
After reading the whole article I would like to suggest some small changes, improvements and additions.

1. Please make the attached questionnaire available so that future researchers can use it in other studies related to this theme in the future, regardless of the context in which they may be inserted. These same authors, using your study will contribute to the increase of your citations;
2. I would like the conclusion to be improved. A work of this quality cannot end with a conclusion that is reduced to a paragraph of 4 lines;
3. Limitations should come after the conclusions and also Future Research Proposals based on this theme should be written;
4. Finally, I think the following papers can be considered for inclusion in your literature review:

Rodrigues, M., Franco, M., Sousa, N., & Silva, R. (2021). COVID 19 and the business management crisis: an empirical study in SMEs. Sustainability, 13(11), 5912.

I hope to have contributed to help you improve a work that is very well done and that will deserve publication soon.

Best Regards

Reviewer 4 Report

Comments and suggestions for authors in attachment

Round 2

Reviewer 1 Report

NA

Reviewer 2 Report

The author has corrected the paper appropriately.

Reviewer 4 Report

The authors have made sufficient modifications.